# NeurIPS should lead scientific consensus on AI policy

**Rishi Bommasani**
Stanford Institute for Human-Centered Artificial Intelligence (HAI)
Stanford University
rishibommasani@gmail.com

## Abstract

Designing wise AI policy is a grand challenge for society. To design such policy, policymakers should place a premium on rigorous evidence and scientific consensus. While several mechanisms exist for evidence generation, and nascent mechanisms tackle evidence synthesis, we identify a complete void on consensus formation. In this position paper, we argue NeurIPS should actively catalyze scientific consensus on AI policy. Beyond identifying the current deficit in consensus formation mechanisms, we argue that NeurIPS is the best option due its strengths and the paucity of compelling alternatives. To make progress, we recommend initial pilots for NeurIPS by distilling lessons from the IPCC's leadership to build scientific consensus on climate policy. We dispel predictable counters that AI researchers disagree too much to achieve consensus and that policy engagement is not the business of NeurIPS. NeurIPS leads AI on many fronts, and it should champion scientific consensus to create higher quality AI policy.

## 1 Introduction

Building good AI policy is important. Governments around the world agree on this, even as they diverge on the specifics across jurisdiction (e.g. mandatory requirements in the EU vs. voluntary frameworks in the US) and administration (e.g. Biden's focus on AI governance vs. Trump's focus on AI dominance). Of course, the policy that society defines for AI as a transformative technology will simultaneously shape the technology's trajectory and mediate the extent to which it produces desirable outcomes for humanity. With this in mind, AI researchers increasingly publish on AI policy.

Good AI policy depends on many factors (e.g. political climate, constituent interests, market incentives, government capacity), but our focus is on two intertwined elements essential for evidence-based policy [Bommasani et al., 2025a]: rigorous evidence and scientific consensus. We define rigorous evidence as information generated via methods (e.g. theory, evaluations, deployments, simulations) that are transparent, replicable, and methodologically sound. We define scientific consensus as a general agreement among experts in a scientific field based on the accumulation of rigorous evidence over time. Rigorous evidence and scientific consensus are both valuable for policy design, but many practicalities complicate their relationship with policy in reality.[1]

While rigorous evidence and scientific consensus are naturally entangled, we disentangle them because they differ significantly in their maturity for AI policymaking. Many processes generate relevant evidence on AI (e.g. academic scholarship, company reports, journalistic inquiry). With that said, certain key evidence, such as the distribution over use cases for foundation models [Bommasani et al., 2023, Stein et al., 2024], remains very minimal. This paradox is made clear at NeurIPS: NeurIPS 2025 is inundated with over 25000 submissions, yet critical information about AI and its

---

[1]History demonstrates that important policy can be set without the consultation of scientists. While we acknowledge evidence and consensus often emerge slowly, we see the better path as aggressively pursuing evidence and consensus so as to avoid this fraught prospect.

39th Conference on Neural Information Processing Systems (NeurIPS 2025) Position Paper Track.

impacts remains unclear. For this reason, evidence synthesis is necessary to extract signal from the noise to better inform policy. While comparatively fewer mechanisms exist for evidence synthesis than evidence generation,[2] the International Scientific Report on AI, led by NeurIPS Advisory Board member Yoshua Bengio and authored by 96 international experts with the endorsement of 30 nations, has emerged as the premier mechanism for gathering policy-relevant evidence [Bengio et al., 2025].

**No bona fide mechanism exists to develop scientific consensus on AI policy and that NeurIPS should fill this void.**[3] NeurIPS is the right site for scientific consensus formation because of its iconic strengths: NeurIPS is unquestionably the leading AI convening in the world with the drawing power to bring the AI community together and has long stood as a beacon for scientific leadership on artificial intelligence. Therefore, NeurIPS should simultaneously serve as the locus for internally organizing the AI community to pursue consensus and externally communicating such consensus in a reputable and trustworthy fashion. While alternatives exist, such as the AI summit series (i.e. the UK AI Safety Summit in 2023, the AI Seoul Summit in 2024, the Paris AI action Summit in 2025), these alternatives often lack the cachet among scientists of NeurIPS and they do not foreground researchers as is necessary to achieve meaningful consensus on complex scientific issues.

Scientific consensus is not the wheelhouse of NeurIPS and, frankly, is something most AI researchers have never actively worked on. After all, as with other scientific communities, our community is highly distributed in our views on many subjects. This may lead some to misunderstand the very intent of "consensus": our objective is not to claim AI researchers can or should agree on everything. Therefore, to guide NeurIPS and the AI community on this path, we provide (i) concrete lessons from the history of the IPCC in the domain of climate science and (ii) tractable pilots to implement our proposal, which include clear opportunities to foreground debate, uncertainty, and deliberation as central to making headway on meaningful and hard-won scientific consensus.

NeurIPS buttresses the development of AI as a field by bringing the AI community together each year as the broad-tent multidisciplinary venue home to many landmark breakthroughs. But NeurIPS is not merely the collection of papers presented at the conference over its history. NeurIPS has been a pioneer on many fronts that reflect core values: reproducibility (the ML reproducibility checklist), ethics (the ethics review process), transparency (the use of OpenReview), diversity (the first Black in AI workshop), interdisciplinarity (the first ML4Health workshop), and infrastructure (the first Datasets & Benchmarks track). In large part due to the stewardship NeurIPS has provided for almost forty years, AI now is no longer the fledgling offbeat research niche of the 1980s, but the marquee technology of the 2020s that demands an all-hands-on-deck approach for achieving good future outcomes. We have a responsibility to continue NeurIPS's longstanding leadership by forging scientific consensus on the big challenges that await society.

## 2 The Problem: Consensus Formation

Developing good AI policy is multi-step process, much of which is quite distant from AI research. We identify three subproblems that, when best-instantiated, involve the research-policy interface: (i) evidence generation, (ii) evidence synthesis, and (iii) scientific consensus.

NeurIPS plays a central role in evidence generation by catalyzing the production of evidence and, weakly, coordinating the focus of evidence production (e.g. the implicit norms in the peer review process that shape what is valued, the explicit tracks named in the call for papers). The NeurIPS peer review process also provides some signal in identifying evidence as rigorous or credible.[4] Other domains have more mature processes for determining which evidence is credible for the purposes of

---

[2]Of course, we should expect and want fewer mechanisms for synthesis than generation, but we currently feel there are too few reputable evidence synthesis mechanisms for AI policy.

[3]This piece was written prior to the formation of the UN Independent International Scientific Panel on AI, which draws inspiration from the UN Intergovernmental Panel on Climate Change as is discussed in this piece. We do not further discuss the UN Independent International Scientific Panel on AI as its exact function still remains unclear and, more importantly, its role may be in jeopardy given remarks delivered by President Trump at the United Nations in September 2025.

[4]Note that the International Scientific Report on AI does not require work to be peer-reviewed to be sufficiently credible for use: instead the report relies on the judgment of its expert authors to select high-quality sources based on (i) originality, (ii) impact, and (iii) and transparency with respect to the methods used, prior work, limitations and opposing views.

policymaking [Cartwright and Hardie, 2012]. For example, in public health, evidence generally takes the form of observational data that is scored using the GRADE system for evidence [Moberg et al., 2018] to determine eligibility for use in evidence-based health policy [Brownson et al., 2009].

To a lesser extent, NeurIPS plays a role in evidence synthesis, to the extent it publishes surveys and meta-analyses. As is the case in other more mature disciplines (e.g. economics), such work is not recognized by the AI community as being of equal prestige as primary research that establishes novel findings. The NeurIPS call for papers makes no mention to surveys or meta-analyses, though several such works have been published at NeurIPS.[5] In the context of evidence-based policy, evidence synthesis not only organizes the vast distributed evidence base, but also makes such evidence cognizable to policymakers. NeurIPS currently plays no such role, though recent initiatives to have "lay summaries" for published work may have the side-effect of making individual papers more legible to policymakers (e.g. legislative staffers working on specific issues).[6]

NeurIPS currently plays no role in the formation of scientific consensus in relation to AI policy. While NeurIPS does facilitate the organization of workshops which may address AI policy or related topics (e.g. the Workshop on Regulatable ML at NeurIPS 2024), these workshops gather AI policy researchers to discuss recent research. Beyond NeurIPS, we are not aware of any established scientific consensus formation process for AI policy. We present arguments in support of NeurIPS claiming this mantle, while acknowledging the deficiencies of other options that make some sense, and may even be (mis)understood as currently serving the desired role. With this said, we emphasize the purpose of this paper is to animate a push for NeurIPS to take on this role, partially due to its distinctive position, and parallel efforts elsewhere may be useful.[7]

## 2.1 Strengths of NeurIPS

We argue that NeurIPS is the best option for driving scientific consensus formation on AI policy because of its unique role both internally and externally. Internal to the AI scientific community, NeurIPS has unparalleled power to bring scientists together. External to the AI scientific community, NeurIPS has unparalleled reputation to represent AI scientists. To reason about these strengths, and the extent to which they are relevant for consensus formation, we draw upon the multidisciplinary literature on consensus in disciplines such as sociology, philosophy, politics, and economics [see Zollman, 2012, Stegenga, 2016, Miller, 2019].

**Internal role.** To meaningfully claim scientific consensus, a consensus formation process should be *legitimate*: the process should reflect expertise (i.e. the participants are scientists) and be inclusive (i.e. scientists can easily participate). NeurIPS clearly brings together leading scientists on AI each year, though many others currently attend NeurIPS (e.g. recruiters, journalists). And attendance to NeurIPS is broadly accessible: NeurIPS 2024 had over 16000 in-person attendees from around the world, spanning a variety of disciplines and sectors. While certain barriers do routinely arise for NeurIPS attendance (e.g. visa issues), NeurIPS is well-positioned to facilitate virtual engagement on consensus due to extensive use of OpenReview already.[8]

**External role.** To meaningfully act upon scientific consensus, scientific consensus should be *credible*: the consensus should be the byproduct of a legitimate process grounded in evidence to reflect expert agreement absent external coercion. Much of this is covered above: if NeurIPS designs a consensus process convincing to AI scientists, this will likely be convincing to the world, and policymakers in particular. Further, the sterling reputation of NeurIPS bolsters this case: NeurIPS is not only the premier AI conference, but is one of the most prestigious venues for research in all of

---

[5]Note that TMLR does explicitly invite "surveys hat draw new connections, highlight trends, and suggest new problems in an area", and even awards recognitions for outstanding survey papers.

[6]Note that the marginal benefit to policy of such summaries is likely minimal: similar quality summaries could be easily produced by today's language models on demand and such generic summaries do not substitute for well-written policy briefs that deeply integrate evidence into policy contexts.

[7]In contrast to other initiatives where parallel work streams are generally beneficial even when redundant, parallel efforts may be detrimental for consensus formation. Namely, consensus formation intrinsically strives for a collective view where possible, whereas parallelism promotes fragmentation and de-synchronization.

[8]In addition, NeurIPS already is sufficiently well-known, which helps to mitigate travel restrictions that are likely to also plague other in-person convenings.

science.[9] External coercion may be a cause for a concern due to entanglement between NeurIPS and the AI industry, which has clear self-interests in the design of AI policy [Casper et al., 2025a].[10] We believe this concern can be managed, much akin to how NeurIPS navigates similar challenges in the conflict-of-interest process for paper review and the oversight of its ethics review process.

## 2.2 Weaknesses of alternatives

We identify two categories of alternatives: those that claim to play some role in scientific consensus, and those that do not claim some role but could plausibly fulfill it. We describe notable options within each category, indicating how they operate and why their structure is unideal for our vision of scientific consensus.

**International Scientific Report on AI.** The International Scientific Report on the Safety of Advanced AI [Bengio et al., 2025], which we abbreviate as the International Scientific Report on AI, was commissioned at the 2023 UK AI Safety Summit and published at the 2025 Paris AI Action Summit. Turing Award winner Yoshua Bengio led the writing group of 96 AI experts, including an international Expert Advisory Panel nominated by 30 countries, the OECD, EU, and UN. The report's self-described primary purpose is evidence synthesis, which includes identifying areas of preexisting scientific consensus. Our position is the report is the best evidence synthesis available for AI policy and this should remain its primary purpose. The report writing process taken many months for a team of over a hundred, including dedicated administrative staff beyond the expert writing team, so it does not make sense to expand its scope to also take on consensus formation, but instead reports can communicate the outcomes of ongoing scientific consensus formation processes.

**AI summit series.** The AI summit series was initiated by the UK government under Prime Minister Rishi Sunak with the 2023 UK AI Safety Summit, followed by the 2024 Seoul AI Summit and the 2025 Paris AI Action Summit. The summits are high-profile convenings of world leaders, industry executives, and AI experts, among others, with some some summits yielding high-level commitments from leading companies and countries on AI. Therefore, the summits build momentum on an international agenda for AI, but the conversations do not reach the acuity needed for substantive work on scientific consensus. Moreover, the summits are inextricably entangled with high-level (geo)politics. Consider the remarks from US vice presidents, leading the American delegations at the summits, in 2023 vs. 2025. In 2023 at the UK AI Action Summit, vice president Kamala Harris stated the US priority was to "consider and address the full spectrum of AI risk: threats to humanity as a whole, as well as threats to individuals, communities, to our institutions, and to our most vulnerable populations" whereas in 2025, Vice President J.D. Vance stated that "I'm not here this morning to talk about AI safety, which was the title of the conference a couple of years ago. I'm here to talk about AI opportunity." Depoliticized scientific consensus will not flourish amidst these strong political undercurrents. Our position is the AI summit series should continue, insofar as there remains broad global support, but they will not work as a site for scientists to develop consensus.

**Other options.** Given we focus on NeurIPS, a natural alternative is other AI conferences (e.g. FAccT, ICML, CVPR, ACL). While each of these venues has their strengths, we believe NeurIPS Pareto-dominates each of them as the site for scientific consensus on AI policy: NeurIPS has the broadest attendance from the many subareas of AI (i.e. more legitimate due to greater inclusion) and often is the most reputable (i.e. more credible). The International Association for Safe and Ethical AI (IASEAI), which recently held its first conference, may hold promise, but the organization lacks the track record of NeurIPS and it is generally premature to understand whats it function will be. Beyond conferences, different advisory groups may provide other types of expert-informed guidance (e.g. the UN High-Level AI Advisory Board), but these bodies often have other roles, and the level of inclusion will generally pale to NeurIPS by design. Finally, a few recent AI policy papers authored by larger groups from different institutions demonstrate inroads towards consensus on specific topics among smaller coalitions of experts [Brundage et al., 2018, Kapoor et al., 2023, Ho et al., 2023, Anderljung et al., 2023, Bengio et al., 2024, Kolt et al., 2024, Kapoor et al., 2024, Longpre et al.,

---

[9]NeurIPS is ranked as $7^{th}$ among all scientific venues by Google Scholar for its h5-index.

[10]Examples include the large presence of industry among conference attendees, which is not limited to industrial researchers; the composition of NeurIPS leadership; and the extensive funding dependence of NeurIPS on the AI industry.

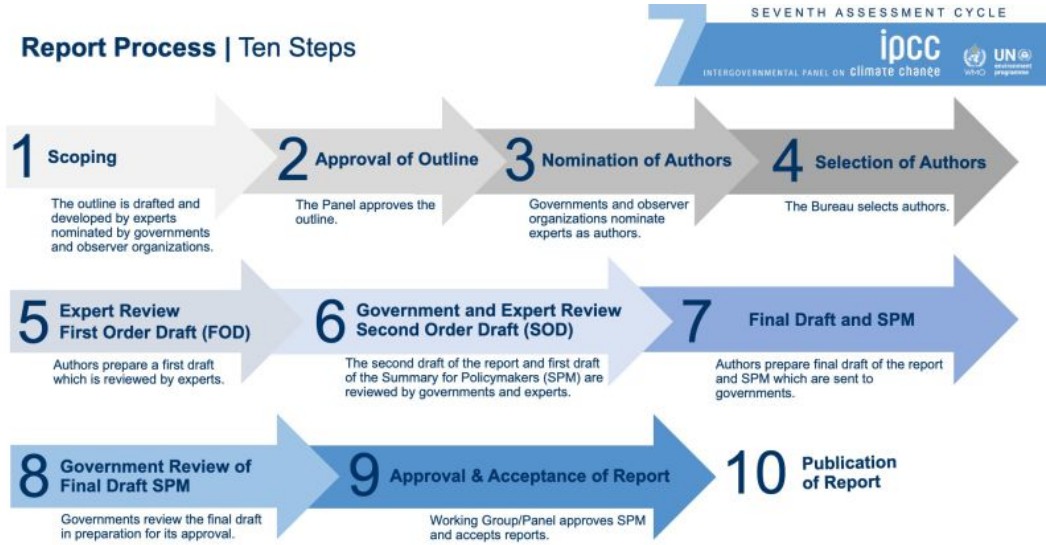

Figure 1: **The IPCC Assessment Report process.** To develop scientific consensus on climate policy, the IPCC conducts an elaborate multi-step process to ensure the legitimacy and credibility of the resulting scientific consensus.

2024, Bommasani et al., 2025a]. These works do not substitute for robust large-scale scientific consensus formation process, but their authors comprise a growing collective that may be inclined to implement the pilots we believe NeurIPS should pursue.

## 3  Guidance on the Process

The demand for consensus in artificial intelligence as a discipline has been limited: while all fields may, in spirit, endeavor to achieve consensus in their pursuit of scientific truth, little concretely depended on such consensus. In contrast, clearly articulated consensus on specific topics would be valuable now for AI policy to prioritize interventions that are technically feasible and supported by research [Guha et al., 2023]. To guide the process, we review the highly respected process of the Intergovernmental Panel on Climate Change (IPCC) to achieve scientific consensus on climate science [De Pryck, 2021]. The resulting lessons yield proposals for pilots that we situate in the context of NeurIPS.

### 3.1  Lessons from the IPCC

The IPCC was established by the United Nations in 1988 to "prepare a comprehensive review and recommendations with respect to the state of knowledge of the science of climate change; the social and economic impact of climate change, and potential response strategies and elements for inclusion in a possible future international convention on climate". The primary IPCC deliverable is the Assessment Report every five years: over the past 37 years, the IPCC has prepared six reports that each have been closely tied to high-level international climate policy (e.g. the Kyoto protocol, which set legally binding emission reduction targets for developed countries; the Paris agreement, which set a target of limiting global warming to well below 2°C). The IPCC has orchestrated broad changes to international practices to materially improve climate outcomes, receiving the 2007 Nobel Peace Prize for "for their efforts to build up and disseminate greater knowledge about man-made climate change, and to lay the foundations for the measures that are needed to counteract such change". We focus our attention on central elements of the IPCC that provide guidance for scientific consensus formation on AI policy with the IPCC process depicted in Figure 1.

**Legitimacy.**  To ensure inclusion, the IPCC process begins with a broad call for nominations, where experts are nominated by governments and organizations with standardized materials (e.g.

CV, relevant publications, field of expertise). Given these nominations, the overseeing Bureau of experts select the authors based on topical expertise, geographic balance, gender balance, institutional diversity, and disciplinary diversity. In this way, the underlying process for selecting the involved scientists comes with strong assurances for both inclusion and expertise.

The authors are organized into three working groups: (i) the physical science basis, (ii) impacts, adaptation and vulnerability, and (iii) the mitigation of climate change. Authors participate in working groups based on their expertise, thereby ensuring any resulting consensus is tightly connected to the expertise of the participants. These working groups iterate on many aspects of the Assessment Reports (e.g. the underlying gathering and synthesis of evidence), but we focus on the consensus formation elements. Namely, the IPCC implements consensus through reasoned agreement rather any form of voting, by having working groups convene repeatedly throughout the year with dedicated opportunities for extended discussions and disagreements. If consensus cannot be achieved, the Report is written with standardized calibrated terminology, tied to concrete likelihood scales, to communicate uncertainty to honestly reflect disagreement without forcing artificial consensus. Reports may explicitly describe alternative interpretations to identify consensus among smaller groups and reports attach comments that reflect dissent from any of the authors or reviewers involved to maximize transparency.

**Credibility.** The credibility of the IPCC process begins with the deep integration of governments in the selection of experts and the determination of the scope of the effort, while maintaining the independence of any scientists involved in the process. On top of this, the IPCC process includes several rounds of external review for each report, with the UN Secretary-General Ban Ki-moon and IPCC Chair Rajendra Pachauri having the InterAcademy Council (IAC) to conduct an independent review of the entire IPCC process in 2010. To minimize the risk of external coercion and conflict of interest, the IPCC publishes a detailed policy: experts are selected independent of political affiliations and cannot be coerced by governments, conflicts of interest must be declared at the onset, and the high-status nature of the process creates reputational penalties for misconduct.

**Impact.** To maximize positive impact, the IPCC takes steps beyond creating the conditions for legitimate and credible scientific consensus to form. To ensure it has the desired type of impact, the IPCC sets clear expectations for its work to avoid misinterpretation. Namely, the IPCC self-describes its work as policy-relevant but not policy-prescriptive, and it deliberately involves governments to ensure buy-in while maintaining boundaries to protect the independence of the scientists involved. To ensure the work is comprehensible to policymakers, each IPCC Assessment report features a Summary for Policymakers (SPM). As the most salient part of the entire report [De Pryck, 2021], each government edits this text on a per-line basis, but scientists retain the ability to veto any changes in the event they do not reflect the scientific consensus. Once endorsed, the SPM content should not be renegotiated in the UN's overarching framework for climate change, thus granting the SPM a "perceived binding force" Riousset et al. [2017]., Overall, the report makes the scientific consensus clear, and presents it in a way useful for policymakers, while neither overstepping beyond its desired scope nor compromising on the underlying scientific consensus.

While we present a broadly positive account for the IPCC process and its impact, we do acknowledge critique that results from the focus on consensus [Oppenheimer et al., 2007]. Namely, to achieve consensus, IPCC reports often prioritize expected outcomes grounded to numerical estimates, since this is where consensus can be achieved. Therefore, such a focus on consensus may serve to exclude or downplay more extreme possibilities [Patt, 1999], which are also important for policymakers to understand. Notably, such concerns about extreme risks [Bengio et al., 2024] are the same areas where we see the greatest disagreement among AI experts Bengio et al. [2025].

### 3.2 Pilots to implement

We propose pilots based on general arguments for their benefits, but the specific next steps that will make the most sense will require input from the NeurIPS community and leadership. The pilots are designed to be low-cost interventions to distinguish whether (i) NeurIPS should further invest to cement its role as the leading site for scientific consensus formation on AI policy or (ii) NeurIPS should not further pursue this activity. Critically, the pilots also address different levels of the NeurIPS process: the working group is an overall year-round initiative, the dedicated track would be tied to paper submissions, and the surveys/debates would be tied to the conference event itself. By operating

at these different levels of granularity, the pilots may have synergies if implemented simultaneously (e.g. the working group could naturally source volunteers to oversee conference events) and enable differing levels of engagement.

**Working group.** NeurIPS could create a standing working group, drawing upon leaders in the community along with researchers who specifically work on AI policy. The working group would provide sustained leadership to shepherd scientific consensus formation, including any events conducted during NeurIPS. Further, while much of the current conceptualization of NeurIPS builds to the conference dates in December, the nature of scientific consensus formation will require more intermittent dialogue throughout the year. Therefore, this group would be responsible for carrying out this work, providing their conclusions in a special session at NeurIPS.

**Dedicated track.** NeurIPS could define a special track or theme for a single-year pilot to promote scholarship on AI policy with an eye towards consensus. In recent years, the conference has explored creating new tracks to support new trends (e.g. the datasets and benchmarks track) with the current position paper track potentially being a natural home for policy scholarship, as we have seen at ICML 2024 and ICML 2025. However, distinct from the existing tracks, this track could promote work that actually forges scientific consensus. For example, such a track could promote surveys of the existing views of scientists on specific issues, meta-analysis of conflicting evidence that may preclude consensus, and new methods like if-then protocols for achieving consensus [Karnofsky, 2024].

**Debates and surveys.** NeurIPS could create a special session during the conference for a spirited debate on current areas where consensus is lacking. While many workshops at NeurIPS have hosted debates over the years, the main program for NeurIPS generally does not feature debates (instead favoring keynotes and panels), but debates can garner better attendance and force the deliberation required to pursue non-obvious consensus. To supplement such a debate, NeurIPS could conduct surveys both prior to and after the conference to understand the latent level of agreement on specific topics within the community. The NLP community conducted a similar metasurvey in 2022 to assess researcher views on "controversial issues, including industry influence in the field, concerns about AGI, and ethics" [Michael et al., 2023]. NeurIPS, in particular, has administered surveys and experiments on the community: the 2014 and 2021 experiments on peer review consistency, the 2021 survey on demographics, and the 2024 survey on language models as author checklist assistants.

## 4 Examples of the Substance

This paper's primary purpose is to argue for the role of NeurIPS in scientific consensus formation, but it is difficult to reason about the merits of this position without understanding what we may attempt to achieve consensus upon. Namely, irrespective of the arguments made throughout this piece, many in the AI community may be wary of wading into more highly politicized topics, especially given current animus towards scientists among some US policymakers. Therefore, we present two topics where scientific consensus is lacking, yet both depend heavily on scientific expertise and often bottleneck policy in practice.

### 4.1 Evaluation selection

Evaluations are the de facto method for understanding the capabilities and limitations of models [Raji et al., 2021, Liang et al., 2023, Weidinger et al., 2025]. In the context of policy [Nelson et al., 2024], evaluations can scope a policy (e.g. a particular entity is regulated in a certain way if their performance on an evaluation warrants scrutiny) or be the substantive obligation of the policy (e.g. a particular entity may be required to run an evaluation and report its results to demonstrate its compliance).[11] In many cases, policymakers would like to interpret evaluations as those that satisfy expert consensus: the EU AI Act requires providers of general-purpose AI models with systemic risk to "perform model evaluation in accordance with standardised protocols and tools reflecting the state of the art". But what is the state of the art for model evaluation? Scientific consensus on the answer

---

[11]As an instantiated example, a policy on cybersecurity risk could require developers of AI agents to (i) implement cybersecurity mitigations if their agents score highly on CyBench [Zhang et al., 2025] or (ii) run and report CyBench scores because their agents pose cybersecurity risk.

would be simultaneously invaluable for policy, and appropriate for NeurIPS to attempt consensus formation on.

What would scientific consensus on state of the art evaluations look like? First, consensus may center elements of scientific rigor: which evaluations demonstrate sufficient validity and reliability for the constructs they aim to measure [Jacobs and Wallach, 2021, Weidinger et al., 2025]? Second, consensus may provide independent measures fo cost: how costly would evaluations be to run and what options exist to reduce costs? Finally, the process of consensus formation may reveal gaps (e.g. the lack of a biosecurity evaluation that currently garners consensus may be informative both for future AI and biosecurity research as well as for policy on AI biosecurity risks). While the full policy will hinge on factors that go beyond science (e.g. the amount of burden to impose, meaning the total cost that can be expected for running state of the art evaluations), scientific consensus on the underlying primitives of which evaluations are trustworthy and how to reason about their costs would be invaluable.

## 4.2 Threshold design

Thresholds are the de facto method for determining which entities are covered by a policy [Bommasani, 2023, Heim and Koessler, 2024]. The most common example in policy is training compute thresholds: the Biden administration's flagship Executive Order on AI required developers to report their red-teaming results for "any model that was trained using a quantity of computing power greater than $10^{26}$ integer or floating-point operations . . .". Naturally, any kind of threshold is tightly connected to the state of science, because the goal is to select viable quantitative metrics that can proxy for the most risky models. The current scientific literature reveals a clear lack of consensus: some argue compute thresholds are desirable if executed well Heim and Koessler [2024], Casper et al. [2025b], others argue compute thresholds should be avoid as poor proxies for risk [Hooker, 2024], and still others argue compute thresholds should not be used alone but they may be viable when combined with other closer proxies of risks [Bommasani, 2023, Nelson et al., 2024, Bommasani et al., 2025b].

What would scientific consensus formation on thresholds reveal? First, many metrics could be candidates for use in thresholding, but relatively few have been seriously considered by policymakers: broad discussion would identify new options (e.g. thresholds related to properties of organizations rather than specific models), including hybrid solutions (e.g. thresholds based jointly on compute and evaluation results) that may yield Pareto improvements. Second, current candidates reflect trade-offs (e.g. training compute can be measured reliably but may be a poor proxy; post-deployment usage statistics can measure risk well but may be difficult to measure), as is clear in the current literature: work towards forming consensus would develops the principles on how to make decisions in spite of these trade-offs (e.g. the relationship between predictive validity of risk and measurement costs). Finally, working towards scientific consensus on thresholds would reveal more deficits in infrastructure: where can policy and research work together to create the tooling required to measure certain quantities that would yield better thresholds?

## 5 Conclusion

We advocate for NeurIPS establishing itself as the leader in driving scientific consensus formation, which can become a powerful primitive for evidence-based AI policy. This paper identifies the deficit in current consensus formation mechanisms, reasons for why NeurIPS outclasses other considered options, and guidance for how to fill the gap. What we are proposing will strike many as unorthodox for NeurIPS, because it is. But we write this position because we believe what we describe is uniquely possible for NeurIPS to achieve and, in doing so, will realize the most fundamental mission of NeurIPS: to steward scientific inquiry on AI to ensure AI produces beneficial outcomes in society.

## 6 Alternative Views

In §2, we discuss the alternative sites for scientific consensus formation, and resolve why we see NeurIPS as the best option. Here, we discuss two stronger alternative views: (i) scientific consensus formation may happen, but it is not in keeping for it to happen via NeurIPS and (ii) consensus formation is not worth attempting, because the AI community is too fragmented.

## 6.1 Consensus formation does not belong at NeurIPS

This view, put bluntly, says "This is not the job of NeurIPS. NeurIPS is a site of science, not policy, and we should not broach this topic." Our perception is a sizable fraction of the NeurIPS community may subscribe to this view due to a general aversion to both policy and politics within computer science and AI research, and a further belief that such topics are beyond the remit of AI conferences like NeurIPS. The current political climate may strengthen this view, given a growing praxis to separate computer science/AI from policy to adopt the posture of dispassionate science.[12]

In response, we raise three points. First, irrespective of whether NeurIPS plays an active or knowing role, NeurIPS is a site of scientific consensus formation. While the specific subject of scientific consensus as the basis for evidence-based AI policy is new, the more general endeavor of scientific consensus around the growing knowledge base of scientific knowledge is fully in keeping with the agenda of NeurIPS (and enmeshed with the concept of peer review as a marker of credibility). Second, AI policy is already clearly within the remit of work submitted to NeurIPS.[13] Further, in our discussion of scientific consensus, we deliberately focus on topics that hinge heavily on scientific expertise (e.g. which evaluations to trust), rather than more general topics that hinge on broader ideological stances (e.g. is it good to regulate AI currently?). Finally, the remit of NeurIPS has never been fixed, and always is expanding: a paper on "participatory and representative human feedback for multicultural alignment" would have been unthinkable as in-scope in 1987, yet won a paper award at last year's NeurIPS. Over its history, NeurIPS has come to span many disciplines (e.g. computer science, neuroscience, cognitive science, statistics), so it is only fitting for true scientific consensus on AI policy that reflects these diverse scientific perspectives to come from NeurIPS.

## 6.2 Consensus is unattainable due to underlying division

This view, put bluntly, says "We cannot forge consensus. NeurIPS brings together AI researchers that disagree too much about fundamental issues". Our perception is there is, indeed, immense disagreement among scientific experts on AI at present, let alone its trajectory for the future. For example, some experts claim AI will entirely displace human labor by 2027, while other experts claim AI will be a very important but nonetheless normal technology.

In response, we raise three points. First, the perception that consensus will be difficult to achieve should not deter us from even attempting it first. NeurIPS, and scientific research in general, is grounded in a tradition of attempting hard problems that many initially dismiss as impossible: the climate scientists who built scientific consensus via the IPCC also vehemently disagreed, and only achieved consensus through sustained effort. Second, consensus can be achieved among those who fundamentally disagree by identifying non-obvious yet valuable common ground. On certain subjects, where experts may significantly disagree (e.g. the relative prioritization of near-term vs. longer-term AI risks), shared primitives like better measurement infrastructure can enjoy consensus [Toner, 2024]. Finally, consensus can be achieved by finding areas of uncertainty and agreeing on procedures to reduce that uncertainty. For example, the International Scientific Report on AI indicates "high-level consensus has emerged that risks posed by greater AI openness should be evaluated in terms of 'marginal' risk", even if experts disagree on the current level of marginal risk.

## Acknowledgments and Disclosure of Funding

I thank Alondra Nelson, Ajeya Cotra, Arvind Narayanan, Dan Ho, Dawn Song, Eric Horvitz, Fei-Fei Li, Gillian Hadfield, Helen Toner, Jack Clark, Jacob Steinhardt, Joelle Pineau, Marietje Schaake, Miles Brundage, Ollie Ilott, Percy Liang, Rob Reich, Sayash Kapoor, Seth Lazar, Stuart Russell, Suresh Venkatasubramanian, Tino Cuéllar, William Isaac, and Zico Kolter for important leadership that shaped my views and led me to write this piece. I especially thank Yoshua Bengio for the special role he has played in building the research-policy interface for AI. I was funded by the Stanford Lieberman fellowship at the time of writing.

---

[12]To exemplify this view, see well-known AI researcher Boaz Barak's piece in the New York Times: `https://www.nytimes.com/2025/05/02/opinion/work-school-classroom-politics-harvard.html`.

[13]For example, the position paper track's call for papers references AI policy topics such as "data legality, copyright, intellectual property, privacy, open-source versus closed-source, and regulation of ML technology (licensing, evaluation, disclosures, post-deployment monitoring, etc.)"

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
