# OpenReview forum: "NeurIPS should lead scientific consensus on AI policy"
_NeurIPS.cc/2025/Position_Paper_Track — NeurIPS 2025 Position Paper Track Oral_

### Official Review · Reviewer_q3L5 · 2025-07-12

**Significance:** 3
**Presentation:** 3
**Rating:** 5
**Confidence:** 4

**Summary:**

This paper argues that the NeurIPS conference should actively take on the role of catalyzing scientific consensus on AI policy. The authors identify a "void" in consensus formation mechanisms, distinct from existing evidence generation and synthesis efforts. They posit that NeurIPS is the ideal institution to fill this void due to its unparalleled convening power and reputation within the AI community. The paper supports its position by drawing lessons from the Intergovernmental Panel on Climate Change (IPCC) as a successful model for building policy-relevant scientific consensus. It proposes concrete pilot programs for NeurIPS—a standing working group, a dedicated paper track, and conference debates/surveys, and provides examples of policy topics (evaluation selection, threshold design) where such consensus is critically needed.

**Strengths:**

The detailed analysis of the IPCC process is highly effective. It serves not just as a vague inspiration but as a concrete model from which actionable lessons on legitimacy, credibility, and impact are drawn. And, the paper moves beyond abstract calls to action by proposing specific, low-cost pilots (working group, track, debates). This makes the position tangible and provides a clear path forward for the community.

**Weaknesses:**

The analogy to the IPCC is powerful, but the paper could more deeply engage with the differences in institutional power. The IPCC is a UN-backed intergovernmental body, which grants it a level of formal authority and political neutrality that an academic conference, even one as prestigious as NeurIPS, lacks. The paper acknowledges industry entanglement but could discuss in more detail how a NeurIPS-led process would maintain credibility against inevitable accusations of being captured by corporate or national interests.

The paper notes the IPCC's "perceived binding force." It would be beneficial to discuss how a NeurIPS consensus could achieve a similar level of influence with policymakers, who may be more inclined to listen to government-sanctioned bodies or industry lobbyists.

**Questions:**

Given that NeurIPS lacks this formal intergovernmental status and is heavily sponsored by industry, how can it realistically replicate the IPCC's perceived independence and credibility in the eyes of global policymakers? Could you elaborate on the specific mechanisms you envision to safeguard this process from both real and perceived conflicts of interest arising from its funding model and the affiliations of its participants?

**Alternative Position:**

Yes, and alternative positions are well-considered and addressed by the argument

**Author Identification:**

No.

**Context:**

3

**Discussion:**

3

**Ethics:**

["NO or VERY MINOR ethics concerns only"]

**Position:**

Yes, the paper argues for or against a position related to machine learning.

**Support:**

4

**Thoroughness:**

4

---

### Official Review · Reviewer_8smc · 2025-08-08

**Significance:** 3
**Presentation:** 3
**Rating:** 6
**Confidence:** 4

**Summary:**

This paper proposes that NeurIPS, as a leading venue in machine learning, should play a central role in guiding scientific consensus and informing AI policy. Drawing on the successful model of the IPCC in climate science, the authors present a framework for AI grounded in three subproblems at the research-policy interface: evidence generation, evidence synthesis, and scientific consensus. They call for actionable pilots such as working groups, structured debates, and survey-based evaluations to foreground deliberation and manage uncertainty. The authors argue that NeurIPS is uniquely positioned to shape credible, rigorous, and democratic consensus-making processes within the AI research community.

**Strengths:**

The paper presents a timely, well-structured, and persuasive case for institutionalizing scientific consensus-building in AI. The analogy to the IPCC is compelling and offers a concrete roadmap. The distinction between internal and external roles is thoughtful, and the authors identify both its global reach and the limitations of existing international AI summits as key context. The proposed implementation pathways, such as working groups, special tracks, and structured debates, are pragmatic.

**Weaknesses:**

While the paper highlights the importance of building consensus, it could provide more detail on how such a consensus would be maintained in the face of AI's fast-moving, commercially-driven landscape. More attention could be given to how the proposed infrastructure would interact with existing governmental, regulatory, or industry-driven efforts.

**Questions:**

Could the authors elaborate on how disagreement and uncertainty would be productively managed within consensus-building? How might NeurIPS coordinate with other international conferences to ensure alignment while avoiding redundancy?

**Alternative Position:**

Yes, and alternative positions are well-considered and addressed by the argument

**Author Identification:**

No.

**Context:**

2

**Discussion:**

3

**Ethics:**

["NO or VERY MINOR ethics concerns only"]

**Position:**

Yes, the paper argues for or against a position related to machine learning.

**Support:**

2

**Thoroughness:**

5

---

### Official Review · Reviewer_nL39 · 2025-08-08

**Significance:** 4
**Presentation:** 3
**Rating:** 8
**Confidence:** 4

**Summary:**

The paper argues that NeurIPS should take a more active role in facilitating scientific consensus formation in AI, drawing lessons from institutions such as the IPCC. It highlights the current incentive structure at NeurIPS, which prioritises novelty over synthesis, and suggests that this leads to fragmented, hard-to-interpret evidence for policymakers and the broader public. The authors propose interventions such as dedicated tracks for surveys, meta-analyses, debates, and consensus reports; working groups; and policy-facing lay summaries. Alternative, less ideal options are considered, and potential implementation challenges are noted.

**Strengths:**

* The paper addresses a policy-relevant issue for the AI research community, connecting scientific consensus formation to effective AI policy design. The motivation is clear, and I mostly agree. The scale of NeurIPS outputs makes it difficult for external stakeholders to interpret results without relying on overhyped findings.
* The call for greater support of surveys and meta-analyses is compelling, with the valuable suggestion that such works provide structured understanding and accessibility for non-experts.
* Alternative options are considered, and the discussion of industry entanglement is relevant. The evaluation of threshold-setting and acknowledgment of related initiatives like lay summaries strengthen the argument.

**Weaknesses:**

* The "Lessons from the IPCC" section seemed somewhat unrelated. It could draw stronger, more explicit connections to what NeurIPS could realistically implement, beyond general inspiration.
* The "Pilots to implement" section could better anticipate obstacle, e.g., the low incentive for high-profile researchers to engage in consensus working groups without prestige benefit like in IPCC, and propose mitigation strategies.
* Evaluation is a good example, but the discussion is very open ended. It would be good if the authors could tie in some of their earlier proposals and make this a “case study” for possible solutions.
* The point that it is a good idea for NeurIPS to have a track encouraging surveys, meta-analyses, etc. seems well reasoned, but touching on organizing the logistics of this would be helpful. While they all help form consensus, they likely have very different review criteria and some have overlap with the Datasets and Benchmarks track and Main track.
* Small mistake in: "The report writing process taken many months"

**Questions:**

* Reviewers tend to reward methodological novelty, while overlooking work that yields key insights into how and why previous approaches perform as they do, especially when such contributions are viewed as “incremental.” Negative or non-confirmatory results, which are a normal and valuable part of the scientific process, are also rarely published. This tends to limit the quality and completeness of the evidence base needed to build scientific consensus. Do the authors have any proposals or extensions to address this?
* Is there a priority order for which pilots should be implemented?

**Alternative Position:**

Yes, and alternative positions are well-considered and addressed by the argument

**Author Identification:**

No.

**Context:**

4

**Discussion:**

4

**Ethics:**

["NO or VERY MINOR ethics concerns only"]

**Position:**

Yes, the paper argues for or against a position related to machine learning.

**Support:**

4

**Thoroughness:**

4

---

### Note · Authors · 2025-08-20

**1-11 Submit Again:**

Definitely yes

**1-1 Submission Process:**

5

**1-2 Next Year:**

Clear expectations of what good positions are and why they are worth publishing at NeurIPS to avoid this being a collection of weird lower-quality work.

**1-3 Future Development:**

Clear expectations

**1-4 Interest:**

["Panel discussions with other position paper authors", "Structured debates on controversial topics"]

**1-5 Thoughtful:**

7

**1-6 Supportive:**

10

**1-7 Technical Aspects Versus Position:**

9

**1-8 Gate Keeping:**

3

**1-9 Camera Ready Changes:**

Not sure yet, depends on reviewer comments?

---

### Meta-Review · Area_Chair_LdLr · 2025-09-07

**Rating:** 7
**Confidence:** 3

**Strengths:**

Summary:

This paper argues an important point about role that NeurIPS (being a premier AI and ML conference) should play in AI policy formation. Authors argue that NeurIPS can take inspiration from international forums like IPCC (for climate change). Authors question the current incentive structure which makes it less than ideal for formulating AI policy. The paper proposes having dedicated forums within the conference in the form of dedicated tracks and working groups.

Strengths:

1. The paper raises an important concern about the role that creators of AI (NeurIPS authors) should play in AI policy formulation.
2. The paper identifies the important role that the NeurIPS community (as creator of the AI technology itself) should play in shaping the AI policy, however it is not doing so currently.
3. The proposal about dedicated forums for this cause is commendable.
4. Further, the paper proposes involvement of industry.
5. Description of IPCC process is nice and gives inspiration.

**Weaknesses:**

1. Though the paper draws inspiration from IPCC, it is not discussed in detail and looks somewhat unrelated.
2. Some of the sections need more elaboration, e.g., evaluation, pilots to implement, section about alternative tracks.
3. It is not clear how the proposed framework interact with existing and future government and industry efforts.

**Questions:**

1. Authors should share more details about implementation order for the pilot.
2. It is not clear how uncertainty would be managed, authors should throw more light on this.
3. Given that NeurIPS is sponsored mainly by industry, so there could be possible biases that could influence policy making, it is not clear how would those be handled?

**Ethics:**

No major ethical concerns reported

**Thoroughness:**

1

---

### Decision · Program_Chairs · 2025-09-26

Accept (Oral)